# Monitoring Multi-Scale Ecological Change and Its Potential Drivers in the Economic Zone of the Tianshan Mountains’ Northern Slopes, Xinjiang, China

**DOI:** 10.3390/ijerph20042844

**Published:** 2023-02-06

**Authors:** Lina Tang, Alimujiang Kasimu, Haitao Ma, Mamattursun Eziz

**Affiliations:** 1School of Geography and Tourism, Xinjiang Normal University, Urumqi 830054, China; 2Research Centre for Urban Development of Silk Road Economic Belt, Xinjiang Normal University, Urumqi 830054, China; 3Xinjiang Key Laboratory of Lake Environment and Resources in Arid Zone, Urumqi 830054, China; 4Key Laboratory of Regional Sustainable Development Modelling, Institute of Geographical Sciences and Natural Resources Research, Chinese Academy of Sciences, Beijing 100101, China

**Keywords:** urban agglomeration on the northern slopes of the Tianshan Mountains, GEE, remote sensing ecological index, geodetector

## Abstract

Accurately capturing the changing patterns of ecological quality in the urban agglomeration on the northern slopes of the Tianshan Mountains (UANSTM) and researching its significant impacts responds to the requirements of high-quality sustainable urban development. In this study, the spatial and temporal distribution patterns of remote sensing ecological index (RSEI) were obtained by normalization and PCA transformation of four basic indicators based on Landsat images. It then employed geographic detectors to analyze the factors that influence ecological change. The result demonstrates that: (1) In the distribution of land use conversions and degrees of human disturbance, built-up land, principally urban land, and agricultural land, represented by dry land, are rising, while the shrinkage of grassland is the most substantial. The degree of human disturbance is increasing overall for glaciers. (2) The overall ecological environment of the northern slopes of Tianshan is relatively poor. Temporally, the ecological quality changes and fluctuates, with an overall rising trend. Spatially, ecological quality is low in the north and south and high in the center, with high values concentrated in the mountains and agriculture and low values in the Gobi and desert. However, on a large scale, the ecological quality of the Urumqi–Changji–Shihezi metropolitan area has worsened dramatically compared to other regions. (3) Driving factor detection showed that LST and NDVI were the most critical influencing factors, with an upward trend in the influence of WET. Typically, LST has the biggest influence on RSEI when interacting with NDVI. In terms of the broader region, the influence of social factors is smaller, but the role of human interference in the built-up area of the oasis city can be found to be more significant at large scales. The study shows that it is necessary to strengthen ecological conservation efforts in the UANSTM region, focusing on the impact of urban and agricultural land expansion on surface temperature and vegetation.

## 1. Introduction

In recent years, China has been pursuing urbanization at an unprecedented rate [1], and the urbanization process has reached a large-scale growth phase. Urbanization, as the most intense human activity on the earth’s surface, is a frontier field of research in the examination of its thresholds and hazards [2]. In parallel with economic development, urbanization is causing or is going to create significant damage and possible risks to the surrounding ecological environment [3] and large-scale ecological disturbances are closely linked to the global carbon cycle and climate change [4]. Therefore, ecological change demands greater attention than ever from the academic community.

As an essential support point of the Silk Road Economic Belt, UANSTM will play a major role in the future as an important national resource processing and storage base and a fundamental engine of urbanization and economic growth in Xinjiang. Under western development policies and socio-economic development, human activities have significantly increased the consumption of natural resources and the burden on the ecological quality (EQ) [5], which makes the EQ of the UANSTM, an already arid zone, even more sensitive and fragile. In previous studies, researchers have explored the ecological development and influencing factors of the region from the perspectives of ecological carrying capacity, prediction of land cover, and landscape ecology [6,7]. The studies have shown that high-quality ecology is concentrated in places such as oasis farmland, and that water use and vegetation distribution have a key influence on regional ecology [8,9]. In UANSTM, scholars have conducted some studies by applying landscape ecology and other tools, but ecological monitoring with remote sensing inversion as a method is still relatively rare.

Today’s scholars have a variety of approaches in the assessment of urban ecology, typically integrating remote sensing techniques with econometric methods or new models. Remote sensing satellites are efficient and objective, precise and quantitative, and have good universality, making them a long-standing choice among scholars [10,11]; calculating vegetation indices [12,13], monitoring surface temperature [14,15], multi-indicator integrated evaluation [16], and other methods are examples of their use. However, either because of single indicators or inconsistent data sources for multiple indicators, this makes it difficult for scholars to adequately analyze complicated realities. The Remote Sensing Ecological Index (RSEI) proposed by Hanqiu X et al. better bridges such a gap [17,18,19]. The four indicators of greenness, humidity, dryness, and heat are uniformly calculated from the remote sensing bands, and the RSEI can be derived after normalization and PCA transformation. This more reliable and comprehensive ecological evaluation method has subsequently been widely employed in regional studies of cities [20,21], mining areas [22,23], wetlands, nature reserves, etc. However, if RSEI is applied to small to medium scale research regions, such as the urban agglomeration on the UANSTM, it may encounter the challenge of calculating enormous amounts of data or giving up precision. Hence, this paper uses Google Earth Engine to overcome this problem. Google Earth Engine (GEE) is a remote sensing cloud computing platform with efficient performance [24,25,26] in simplified pre-processing [27,28], cloud removal [29,30], and small-scale studies [31,32].

Based on the GEE, this paper will use Landsat remote sensing imagery to calculate remote sensing ecological indices, visualize ecological spatio-temporal patterns, and explore the drivers of the ecological environment in the UANSTM, in a study that meets the requirements of high-quality regional development. The ecological environment in arid zones is fragile and sensitive, and the ecological environment and its influencing factors vary at different scales. Therefore, this study monitors ecological changes and their influencing factors at different scales in the study area for the first time to provide a reference for regional sustainable development.

## 2. Materials and Methods

### 2.1. Study Region

The UANSTM is in the southern section of the Junggar Basin, with geographical coordinates 42°78′~45°59′ N and 84°33′~90°32′ E. The Tianshan North Slope Economic Zone includes Urumqi City, Changji Hui Autonomous Prefecture, Turpan City, Wujiaqu City, Karamay City, Shihezi City, Wujiaqu City, Kuitun City, Huyanghe City, Tacheng Prefecture (including only Wusu City and Shawan County, the same below), and Corps and regiments (Figure 1), with a land area of about 1.9 × 10^5^ km^2^, accounting for 12% of the total area of Xinjiang. The climate of the research region is temperate continental, with scarce rainfall, ample sunlight, significant evaporation, and huge daily and yearly temperature fluctuations. With the Turpan Basin to the south and the Gurbantunggut Desert to the north, the UANSTM is distinguished by diverse landscapes, including the Gobi, the desert, the mountains, and the basin. The overall topography is high in the middle and low in the north and south.

The UANSTM is the largest, fastest growing, and most industrialized region in Xinjiang. It is one of the 19 city clusters that the State promoted during the 14th Five-Year Plan period and is also the only city cluster involved in the construction of the two crucial border areas and the Silk Road Economic Belt. The totality of Xinjiang plays a radiating and driving function.

### 2.2. Data Sources and Pre-Processing

#### 2.2.1. RSEI Data

The main data for the calculation of RSEI in this research, Landsat TM/ETM+/OLI remote sensing images, were provided by the GEE platform (Table 1). Landsat has a higher spatial resolution than MODIS and AVHRR satellites, which enables better observation of vegetation phenology during the growing season [33,34]. Compared with Sentinel and other high-resolution images, Landsat has the advantage of obtaining uniform and long-time series images. The research selected remote sensing images with less than 30% cloudiness from June to September each year from SR products (Surface Reflectance, surface reflectance products corrected for radiation and atmosphere) and applied a masking algorithm based on the data quality assessment band (QA). Since bodies of water affect the principal component loadings of the RSEI, the global surface water data from the GEE platform was used for masking; 440 images were then extracted from the median, which are the pre-processed images available in this paper. After completing the calculation of RSEI, it was resampled to 100 m for export and spatial analysis in the local ArcGIS software.

#### 2.2.2. Other Data

The land use/land cover (LULC) data in this research were selected from five products at five-year intervals from 2000 to 2020 (Table 2). Ecological change drivers include model factors, topographic factors, climatic factors, and social factors. Model factors are obtained from the RSEI calculation process; topographic factors are selected from DEM data; climatic factors are selected from annual average temperature and precipitation data, social factors include population spatialization grid data, night light data, and GDP grid data.

Among them, the primary and secondary land use comparison catalog is as follows (The LULC data’s accuracy was checked using the regional ENVI 5.6 program and the verification results showed an overall accuracy of 95.83% with a Kappa coefficient of 0.95. Table 3). LULC types for 2020 were sampled at GEE based on Sentinel data and Google Earth imagery. The LULC data’s accuracy was checked using the regional ENVI 5.6 program, and the verification results showed an overall accuracy of 95.83% with a Kappa coefficient of 0.95 (Figure A1).

### 2.3. Research Method

The idea of this research is shown in Figure 2. Firstly, preprocessing and RSEI index calculations are carried out in the GEE platform, while LULC and human interference (HI) are analyzed, and finally, the influencing factors of the RSEI are analyzed.

#### 2.3.1. Degree of Human Interference

In 2010, Ailian C et al. proposed their corresponding disturbance index based on a questionnaire and expert discriminant methods to classify LULC according to the existing LULC types [35] and WJ Song et al. made some adjustments in their study of this region of UANSTM [36]. In this paper, based on the study of WJ Song et al., we further classify the LULC data of the second level to obtain the following comparison table (Table 4).

#### 2.3.2. Remote Sensing Ecological Index

The GEE platform was used to directly access the Landsat surface reflectance dataset to filter cloud volumes and study range, and to use a masking algorithm for clouds in the image to select images from June to September and extract their median values. As large bodies of water can make large errors in the RSEI, a water body mask was performed and the four indicators of the RSEI were calculated.

RSEI is composed of WET, NDVI, LST, and NDBSI (Table 5), where WET represents the humidity of the region and is derived from the tassel cap transformation. The greenness, humidity, and brightness components of the tassel cap transformation have been widely used in ecological monitoring studies [37] and the humidity component is closely related to soil moisture. Normalized difference vegetation index (NDVI) represents the vegetation greenness of the region and is the most widely used vegetation index. Land surface temperature (LST) represents the surface temperature and is calculated based on the Landsat user manual model and the surface temperature of the parameters revised by Chander et al. [38]. Soil drying has a negative effect on regional ecology and urban built-up areas can also contribute to soil drying [39]. Normalized Difference Built-up and Soil Index (NDBSI) combines the soil index (SI) and the building index (IBI) to represent regional dryness.

The four indicators of the RSEI do not have a uniform scale, and to reduce the impact of this difference, the four indicators need to be normalized. Through the normalization process, the four indicators no longer have magnitudes, and the data are normalized in the range of [0, 1], which facilitates uniform data quality during the principal component transformation.
(9)$$N_{t}=\frac{I-I_{min}}{I_{max}-I_{min}}$$
where $N_{t}$ is the normalized index, *I* is the original index, $I_{min}$ is the index’s minimum value, and $I_{max}$ is its maximum value.

The principal component analysis is a dimensionality reduction method that concentrates most indicators in a small number of principal components. This method does not require manual setting of weights and can avoid weight bias of subjective influence. The normalized datasets were then subjected to principal component analysis to compress the amount of information, and the first principal component was selected as the RSEI index, with higher RSEI values indicating better ecological conditions.

#### 2.3.3. Geodetectors

Geodetectors are a tool for applying spatial statistics to analyze the pattern of spatial heterogeneity of geographic quantities and are often used to explore the magnitude of the weight of influence of independent variables on dependent variables and the interactions between influencing factors [40,41,42]. The Geodetector can perform four types of geodetection (Table 6, The model and equation were proposed by Wang et al. [40]). Two of these are used in this paper: factor detection and interaction detection.

The factor detector is used to interpret the extent of the effect of various influential factors on the dependent variable. This study used a factor detector to measure the contribution of different impact factors to RSEI, where a larger value of *q* indicates a greater effect of this independent variable on RSEI.

The interaction detector quantifies the effect of the interaction between multiple influencing variables on the dependent variable and can disclose the extent to which the influence of two factors is strengthened or diminished when they operate on the dependent variable simultaneously. The *q*-value obtained by the interaction detector indicates the influence of the factor in the joint action, and the comparison with the *q*-value of the single factor can obtain the relationship between the enhancement or weakening of the factor in the joint action.

## 3. Results

### 3.1. Analysis of HI

HI reflects the change in the degree of ecological disturbance by anthropogenic activities under the change in land cover. In the following, the spatial and temporal changes of disturbance degree will be reflected from both LULC and HI perspectives.

#### 3.1.1. Land Use Change

Figure 3 indicates that the rapid urbanization process over the past 20 years has led to a noticeable growth in building land, largely urban land, and agricultural land, represented by dry land. The most significant gain in agricultural land was 7195 km^2^, with an average yearly increase of 359.75 km^2^ and a progressive tendency towards continuous patches. Of the growth of land for construction, Urumqi, Shihezi, and Kuitun have experienced the most considerable expansion. Grassland has suffered the biggest loss, declining by 3238 km^2^, or an average of 161.9 km^2^ each year. Despite the gradual growth of the impact of human activity, natural LULC remains dominant. Unutilized land has the largest share, accounting for 50% of the area of all LULC.

A LULC-type conversion proportional chord diagram was produced based on the change in LULC type; mapping the process of conversion to itself was removed the spatial interconversion of land was highlighted. In the period 2000–2005, many natural LULC types, such as grassland, woodland, water, and bare soil, were converted to agricultural land, with a small proportion converted to urban areas; from 2005 to 2010, agricultural land expanded further. The trend of urbanization is evident from 2010 to 2015, accounting for a relatively large proportion of the interconversion of LULC. The expansion of agricultural land slows down during the period 2015–2020, and the conversion of diverse LULC types to grassland is the main direction of spatial movement of land during this period. In terms of the overall change over the 20 years, the conversion of each LULC type to grassland, represented by unutilized land, is the direction of flow that accounts for the largest share, followed by the conversion of each type to agricultural land, forest and water areas being exported outwards, and towns being imported inwards.

#### 3.1.2. Degree of HI

In the distribution pattern of anthropogenic disturbances in the UANSTM from 2000 to 2020, the proportion of grade 2 anthropogenic disturbances is always the largest, stable between 50% and 55% (Figure 4). Grade 6 and grade 7 anthropogenic disturbances were previously the smallest, however, they both increased with urbanization. Grade 7 rose higher than grade 6, which shows that urban expansion is rapid and greater than township expansion. grade 5 also expanded extremely rapidly, caused by the growth of agricultural land. Grade 5 is also expanding very fast, caused by the increase in agricultural land. It is worth noting that the fraction of anthropogenic disturbances at grade 1 has shrunk significantly, and glaciers are diminishing rapidly.

### 3.2. Spatial and Temporal Variation of RSEI

#### 3.2.1. Ecological Quality of the Overall Region

The RSEI values are classified into five grades: poor, fair, moderate, good, and excellent (According to Xu’s study [17], 0.2 interval between each stage; Figure 5). The overall EQ of the UANSTM is at a fair level, with the mean of the 21-year RSEI raster image being 0.37. The mean value of the 21-year RSEI raster image is 0.37, with area of 1.32 × 10^5^ km^2^ in the ‘fair’ category, accounting for 67.93% of the total region, representing the general ecological condition of the research area (Figure 5g). The region of poor EQ is mainly in Turpan City, Karamay City, and the northern part of Changji Hui Autonomous Prefecture, with the ‘fair’ and ‘poor’ grades in Turpan City and Karamay City accounting for over 80% of the total, and the corresponding landscape types are mainly the Gobi and the desert. The mountain–Gobi–oasis pattern is unique to the region, with a certain transition zone between mountainous woodland and oasis farmland and bare land, where most of the ‘moderate’ ecological class areas are located, accounting for 22.33% of the total region, the second largest ecological class. Areas with ‘good’ EQ are mainly located in mountainous woodlands or oases, and areas with ‘good’ EQ are primarily concentrated in the core of woodlands or oases, with the overall proportion of these two areas being relatively small at 9.34%. The common places are in the cities of Shihezi City and Huyanghe City.

On the time scale, the RSEI values fluctuate considerably, with an overall upward trend (Figure 6). There was a ‘fair’ ecological rating for most of the period 2000–2020, with values fluctuating between 0.28 and 0.38; a few years were in the ‘medium’ ecological class, as was the case in 2000, 2014, 2018, and 2019, with RSEI values of 0.48, 0.49, 0.44, and 0.50, respectively. In addition, four years had RSEIs at the ‘fair’ and ‘moderate’ thresholds.

#### 3.2.2. Ecological Quality of Localized Areas

From the perspective of temporal changes in administrative units, the state of ecological change differs from region to region (Figure 7). The city of Urumqi shows a trend of deterioration followed by optimization around 2005, mainly in the ‘fair’ and ‘medium’ grades, while the cities of Karamay, Turpan, Changji Hui Autonomous Prefecture, Ili Kazakh Autonomous Prefecture, Tacheng, and Huyanghe are in a similar position. The situation in Shihezi and Wujiaqu is unique in that the ecological value of these two locations was among the highest in the entire study area, but the area in the ‘fair’ category has increased in recent years, which is a phenomenon worthy of attention.

#### 3.2.3. Ecological Quality of a Typical Region

The RSEI distribution for each decade was spatially overlaid to estimate the ecological changes in the UANSTM over each decade (Figure 8). Large regions of overall regional change were found to be concentrated in unused areas such as the desert and Gobi, with smaller portions and high fragmentation in oasis cities and the Tianshan Mountains’ forest. One of the characteristics of the UANSTM is that natural LULC patterns account for a fairly large proportion of the region, while oasis cities, where human activities are common, account for a relatively minor fraction. A more in-depth assessment of the usual areas of significant change picked from the RSEI change map (referred to as Typical Region) demonstrates that oasis cities (especially areas with rapid urban expansion) have small footprints but large ecological changes, often below −2 or above 2 levels.

### 3.3. RSEI Impact Factor Geographic Detection

To further investigate the influencing factors of ecological and environmental quality in the UANSTM, model factors: LST (X1), NDBSI (X2), NDVI (X3), WET (X4), terrain factors: elevation (X5); climate factors: temperature (X6), precipitation (X7); social factors: population (X8), night lighting data (X9), human interference (X10), and GDP (X11), are used as independent variables, and RSEI as dependent variables, and geographical probes are used to reveal the weights and interaction effects of the influencing factors of RSEI.

#### 3.3.1. Factor Detection

Firstly, the numerical quantities of each driving factor for the five-time points from 2000 to 2020 were discretized by applying the natural breakpoint method in a cycle every five years and transformed into five types of type quantity. This was followed by the construction of a 2.5 km · 2.5 km fishing grid for the study area and the spatial connection of the independent and dependent variables, which allowed the type values corresponding to the spatial location of each grid to be detected, and the explanatory power and significance tables of the factors per cycle were obtained. The spatial distribution of the factors was derived by taking the median value of the 5-year raster of factors (Figure 9).

A higher q statistic suggests a stronger impact for the dependent variable. Comparing the strength of the factor influence for the region as a whole and a typical region, the strength of the impact is found to vary considerably at different scales (Figure 10). At the scale of the UANSTM, model factor > climate factor > topography factor > social factor, X1 and X3 have the greatest influence, whereas X8, X9, and X11 have a weak influence. This shows that surface temperature and vegetation greenness have a substantial impact on the overall ecological environment of the region and that social factors other than HI have a minor effect. Except for X11 in 2000 and X8 in 2020, the *p*-values for all factors are less than 0.01, which implies that the vast majority of the explanatory results for the impact factors are reliable.

At the scale of the typical region, the differentiation between the components is more obvious, with the effects of X2 and X3 being the most prominent, the effects of X1, X6, and X7 decreasing, and the effects of X10 being relatively unaltered. This indicates that at the scale of the metropolitan area, the effects of surface temperature, topography factor, and climate factors weaken, vegetation greenness and regional dryness have the most significant effects, and the effects of a social factor do not differ significantly from those of the region as a whole. Apart from X9 for 2000 and 2010 and X8 for 2020, the *p*-values for all factors are also less than 0.01, demonstrating that factor detection is also reliable at this scale.

#### 3.3.2. Interaction Detection

The findings of factor interaction detection of the overall area demonstrate that most of the factor interaction detection showed a two-factor enhancement (Figure 11). In 2000, the strongest effect was detected when X1 interacted with X3, achieving an effect value of 0.78, indicating the strongest effect on RSEI when surface temperature and greenness were combined; in 2010 and 2020 the situation is the same as in 2000. Collectively, LST has the strongest effect on RSEI when interacting with NDVI, suggesting that surface temperature and vegetation greenness are two key factors that play an important role in EQ which is stable over time in the study area, highlighting the ecological characteristics of the arid zone of northwest China.

The findings of the interaction detection for the typical region show that X2 and X3 have the largest influence on the RSEI in 2000 with a *q*-value of 0.93 and that X1 and X3 have the greatest influence in 2010 and 2020 with *q*-values of 0.929 and 0.881. This indicates that at the scale of the metropolitan area, the interaction between vegetation greenness and dryness has a significant influence on the region, and that although the LST had a very significant effect on the study area when it interacted with vegetation greenness, the influence of the single factor was not high.

This study shows the spatial and temporal layout of ecology from 2000 to 2020 by building an RSEI model and explores the influence of impact factors on RSEI in different scale contexts and monitors the regional ecological changes in response to the requirements of high-quality sustainable development.

## 4. Discussion

### 4.1. Advantages of Building an RSEI Model Using GEE

The research is based on the GEE platform to rapidly process remote sensing data at a large scale objectively and conveniently, providing a basis for an accurate quantitative assessment of the EQ status of the study region [43,44]. In the past, studies have often been limited to large scale due to their large data sizes, and the limited computing power of offline hardware has led to the use of a few remote sensing images with low cloudiness, thus constraining the performance of other image cloud-free areas for RSEI within the same seasonal period [45,46]. This paper relies on the powerful cloud computing capability of Google Earth Engine to extract median values from all remote sensing images with less than 30% cloud cover from June to September (vegetation growth period), making full use of the available image elements during the growth period, thus increasing the utilization of image elements and highlighting the advantages of the cloud platform in large-scale long-time series studies [47,48].

### 4.2. Spatial and Temporal Evolutionary Characteristics of EQ in the UANSTM

The distribution of changes in RSEI in the UANSTM demonstrates that ecological changes are sensitive, notably in the bare soil areas of northern Changji and Turpan City [49]. These places are sparsely vegetated and the EQ much depends on changes in surface temperature and therefore has a high degree of uncertainty and fluctuation, a feature that differs considerably from other areas of China’s humid and semi-humid zones and is caused by the overall arid climate of Xinjiang [50]. The ecological changes in the oasis cities show the particular impacts of human activities on the EQ, with the areas where Urumqi meets Changji and Shihezi often experiencing −2 and greater degrees of ecological degradation at five-year intervals of change, and a range of more severe degradation in the city of Karamay from 2015–2020 [51,52,53]. The changes over 20 years reveal that the ecological degradation of the city of Shihezi, the location where Urumqi and Changji meet, is relatively visible, which is related to the policy of Urumqi–Changji integration and the economic development of Shihezi. The core of the UANSTM is the Urumqi–Changji metropolitan area, while the city of Shihezi is spatially close to this metropolitan area, so the agglomeration circle they form is the economic engine of the UANSTM [54,55,56]. The ecological degradation of these areas is an inevitable result of rapid economic expansion.

On a temporal scale, the ecological environment of the UANSTM first tends to deteriorate over a 20-year period, which may be related to the western development policy [57,58] and the continuous expansion of agricultural land, while the middle and later stages show an optimization trend amidst fluctuations and the EQ of the Urumqi-Changji–Shihezi metropolitan area, a typical area of economic development, is also changing for the better thanks to the policy of returning farmland to the forest [59] and grazing land to grass. Both agriculture and grassland in the study area contribute considerably to the natural ecosystem. Cropland is better adapted to climatic conditions than grassland, but the uncontrolled expansion of cropland can lead to a shortage of water resources in the region, so balancing the relationship between grassland, agricultural land, and water is the key to ecological conservation in the UANSTM [60,61].

### 4.3. The Role of the Different Drivers

Geo-detecting reveals that LST and NDVI are the key elements impacting the ecosystem of UANSTM. A comparison with the LST time series study by X. Zhang et al. revealed that changes in LST in summer and autumn had a negative correlation with RSEI, with LST reaching its highest value in recent years almost simultaneously in summer and autumn of 2006, and RSEI in that year being the lowest value in recent years [62]. EQ of the UANSTM has been improving amidst fluctuations with the increase in the intensity of reforestation projects in recent years and the heightened focus on the construction of ecological civilization in the 14th Five-Year Plan [63]. The results also show that the role of social factors is much weaker than the influence of natural factors. However, it should be noted that the UANSTM is a vast area, of which the area of human activity is only a small part, so when the spatial scale is changed to a large scale, the function of social variables represented by an anthropogenic disturbance on RSEI is considerably enhanced in comparison with the region as a whole, while the role of the influence of anthropogenic disturbance has tended to increase over time in recent years [64].

### 4.4. Research Limitations and Future Work

NDVI plays an important role in the construction of RSEI for vegetation and ecological monitoring [65,66], and it has been widely used in previous RSEI studies [30,37]. However, the limitations of NDVI vary depending on the context of the study area. Cloudiness and topographic limitations cause NDVI to suffer from some atmospheric and saturation problems in the high-density vegetation-covered Tianshan mountain’s forest area [67,68,69], where EVI performs better. While NDVI has some limitations in sensitivity to soil background in low-density vegetation areas, SAVI is superior [70,71,72]. As a result, given the study area’s unique vegetation conditions, ranging from forest to desert, the NDVI selected for this study is relatively universal. One of the future work priorities could be the creation of a new vegetation index to supplement the limitations of NDVI under different conditions. In the preprocessing process, the method of cloud cover removal can be further improved, such as linear interpolation and Savitzky–Golay filter method [73,74].

The spatial and temporal trends of ecological changes are understood in this paper as a qualitative study, and the next step can be to start from the quantitative study of the trends to obtain more detailed simulations and predictions of ecological trends [75]. Future work can have a more in-depth exploration of ecological change. The LST and NDVI mentioned in the paper are two key factors in the region, the LST is influenced by natural factors and the NDVI is closely related to human activities, and the UANSTM has typical human activities such as arable land expansion and grazing, and how these activities affect the regional ecology through their effects on key factors is the next focus [76,77]. In addition, this paper initially explores the differences in ecological evolution at different scales, and the next step can be to further quantify the scale effects and explore the changes in EQ and its influencing factors under a specific differential search radius [78].

## 5. Conclusions

This paper derives the distribution of ecological land and the year-by-year RSEI based on LULC data and Landsat remote sensing images from the GEE cloud platform and analyzes the spatial and temporal variation of the RSEI and its response to ecological space, drawing the following conclusions:

(1) In the distribution of LULC conversions and degrees of human disturbance, built-up land, principally urban land, and agricultural land, represented by dry land, are rising, with an average yearly increase of 359.75 km^2^, while the shrinkage of grassland is the most substantial, with an average of 161.9 km^2^ each year. The degree of human disturbance is increasing overall, with permanent glaciers, which are virtually free from human disturbance, diminishing, while disturbances at levels 6 and 7, signifying urbanization, are increasing.

(2) The ecological environment of the UANSTM is generally at a fair level according to a study of the spatial distribution of RSEI. The mean value of the 21-year RSEI raster image was 0.37, and the proportion of the ‘fair’ grade area reached 67.93%. Regionally, as a whole, high values of RSEI are concentrated in the woodlands and oasis farmlands on the UANSTM, while low values are concentrated in the northern part of the Changji Hui Autonomous Prefecture and Turpan City; however, on a large scale, the ecology of the Urumqi–Changji–Shihezi metropolitan area, a typical area of economic development, is degrading.

(3) The results of the geodetection demonstrate that LST and NDVI are the most essential influencing elements on the ecology of the region, and that LST and NDVI have the greatest influence on RSEI when they interact. For the region as a whole, social variables play a lesser role, while social elements, dominated by the degree of HI, play a more substantial role at small and medium scales. LST has a stronger effect on the broader region than on the urban area. Future research will concentrate on improving indicators, further quantifying research methods, as well as the ecological impact processes of typical human activities.

## Figures and Tables

**Figure 1 ijerph-20-02844-f001:**
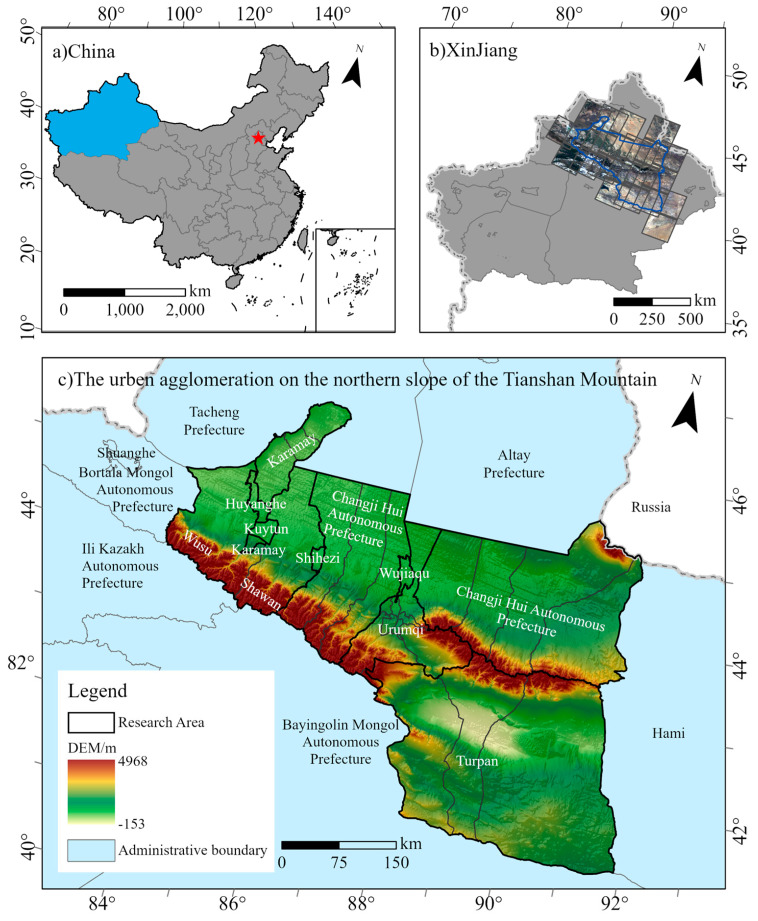
Location of the study region.

**Figure 2 ijerph-20-02844-f002:**
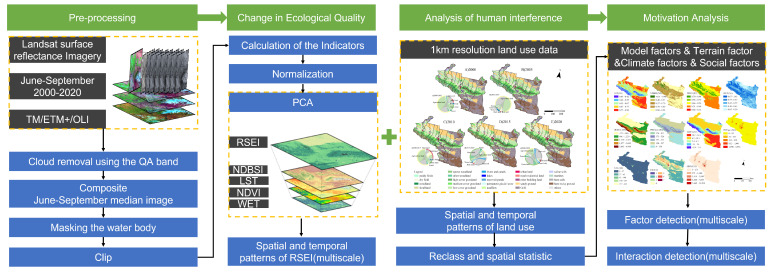
Research framework.

**Figure 3 ijerph-20-02844-f003:**
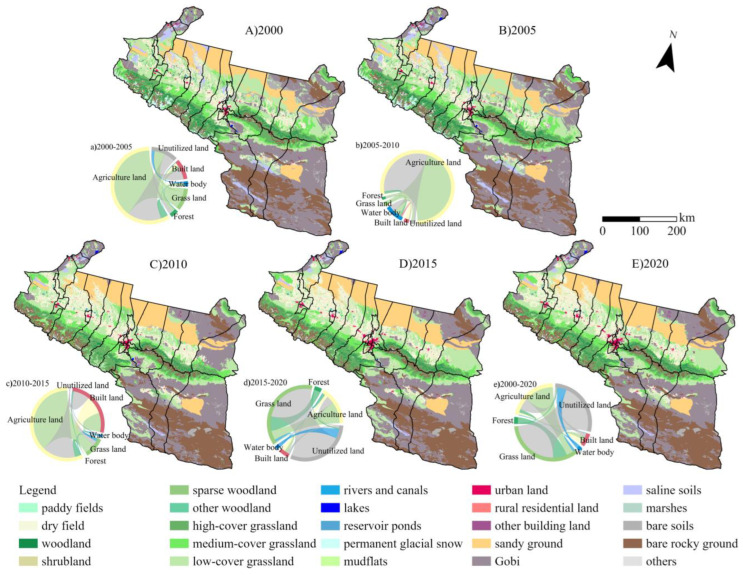
Level 2 LULC type map and proportional chord chart of level1 LULC type transformation in the UANSTM.

**Figure 4 ijerph-20-02844-f004:**
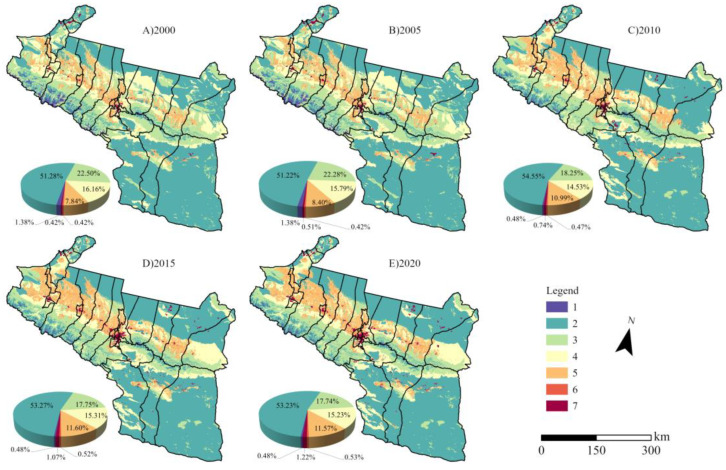
Spatial distribution and statistics of HI per 5 years.

**Figure 5 ijerph-20-02844-f005:**
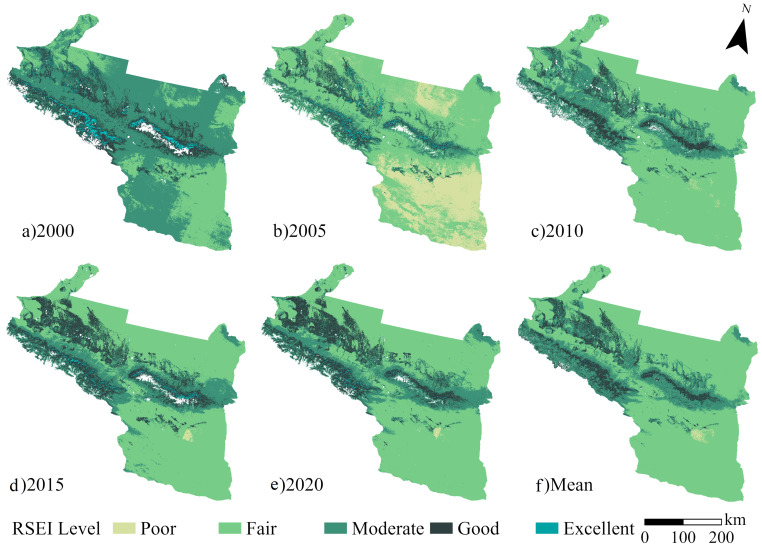
Spatial distribution of RSEIs from 2000–2020.

**Figure 6 ijerph-20-02844-f006:**
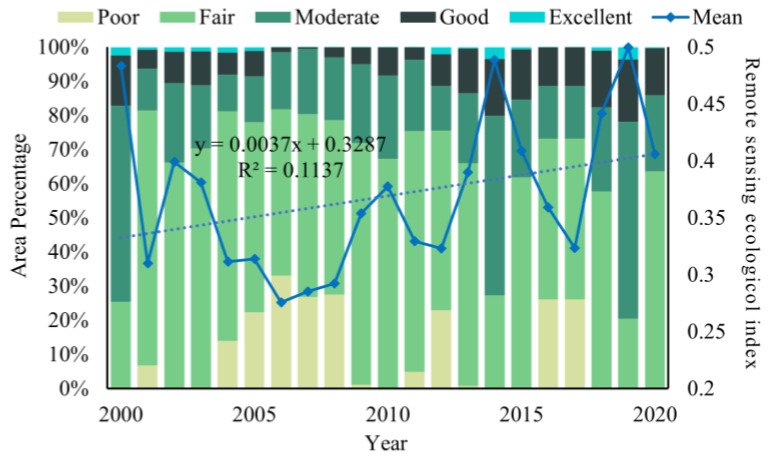
Mean plot on the UANSTM.

**Figure 7 ijerph-20-02844-f007:**
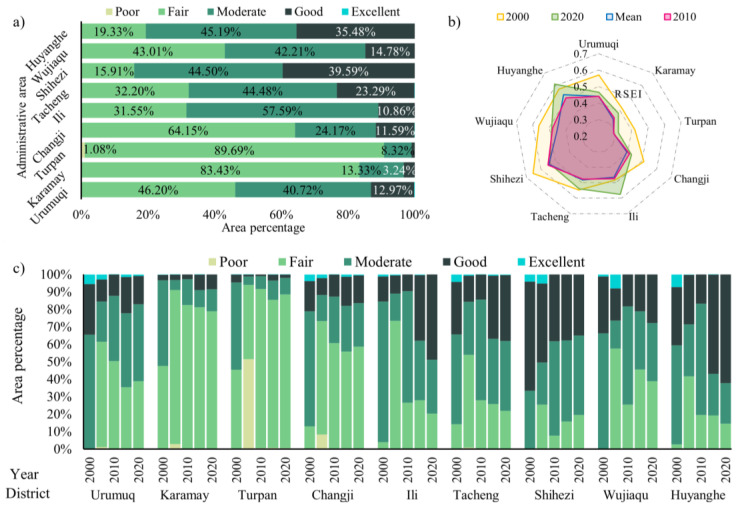
Ecological quality of localized areas. (**a**) Distribution of multi-year RSEI means in different administrative units. (**b**) Annual RSEI for different administrative units. (**c**) Corresponding proportion of RSEI grades and administrative regions of the UANSTM.

**Figure 8 ijerph-20-02844-f008:**
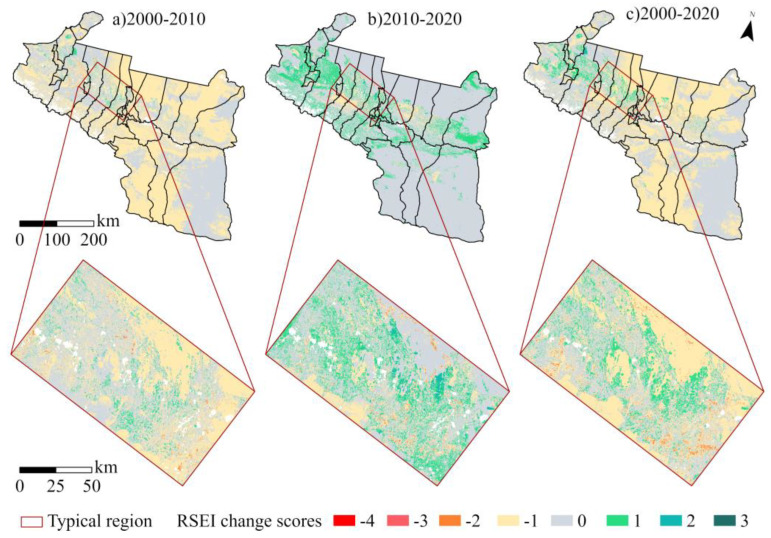
RSEI cycle variation mapping.

**Figure 9 ijerph-20-02844-f009:**
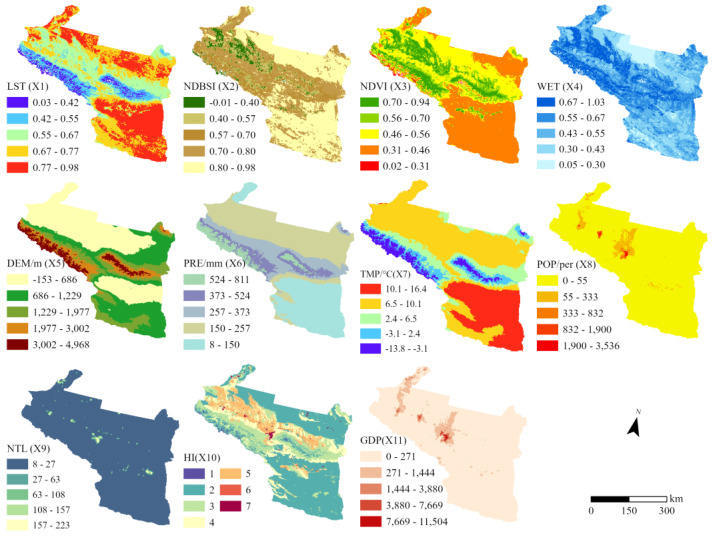
The 5-year median distribution of the factors.

**Figure 10 ijerph-20-02844-f010:**
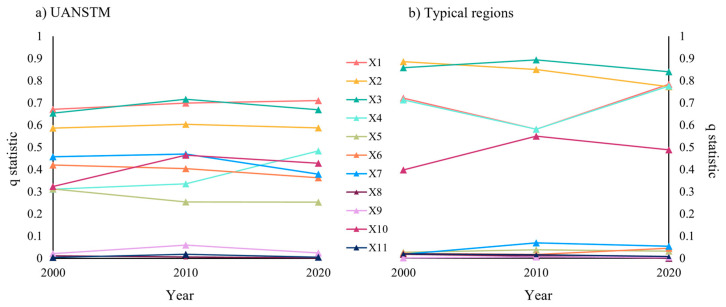
Folding graph of *q* statistic at different scales.

**Figure 11 ijerph-20-02844-f011:**
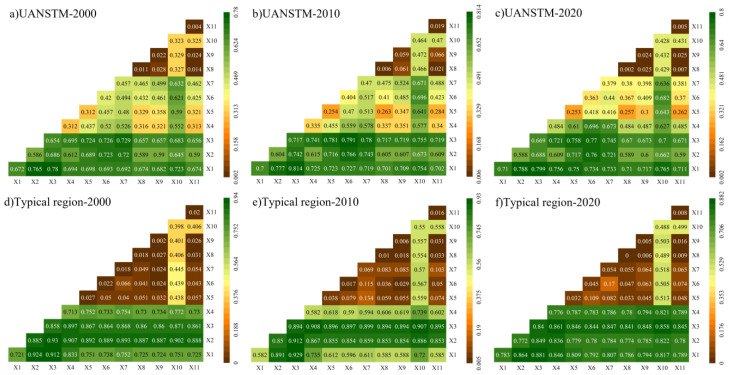
RSEI impact factors results of interaction detection.

**Table 1 ijerph-20-02844-t001:** Basic information on satellite products.

Satellite Sensor	Period	Temporal Resolution	Spatial Resolution
TM	2006–2011	16	30
ETM+	2001–2005&2012	16	30
OLI	2013–2020	16	30

Between 2001 and 2005, images in the study area could not be fully rendered using Landsat5 (TM), so Landsat7 (ETM+) was chosen.

**Table 2 ijerph-20-02844-t002:** Other data sources and periods.

Data Name	Period	Data Sources
DEM	-	Chinese Academy of Sciences Resource and Environmental Science and Data Centre https://www.resdc.cn, accessed on 1 February 2022
LULC	2000–2020 at 5-year intervals	
Temperature, precipitation, and GDP	2000–2015 at 5-year intervals	
	2020	National Earth System Science Data Center https://www.resdc.cn, accessed on 1 February 2022
Night light data	2000–2015 at 5-year intervals	National Oceanic and Atmospheric Administration https://www.ngdc.noaa.gov, accessed on 1 February 2022

**Table 3 ijerph-20-02844-t003:** LULC data level 1 and level 2 comparison table.

Level 1	Level 2
Agricultural land	Dry field, paddy fields
Forest	Woodland, shrubland, sparse woodland, other woodlands
Grassland	High-, medium-, and low-cover grassland
Water body	Rivers and canals, lakes, reservoir ponds, glaciers, mudflats
Built land	Rural residential land, urban land, other building lands
Unutilized land	Bare rocky ground, Gobi, sandy ground, bare soils, saline soils, marshes

**Table 4 ijerph-20-02844-t004:** Degree of human interference vs. LULC comparison table.

Level	Degree of Human Interference	Type of Land Use
1	Virtually immune to anthropogenic influence	Permanent glacial snow
2	Slight anthropogenic influence	Woodland, shrubland, mudflats, marshes, bare rocky ground, Gobi, sandy ground
3	Moderate anthropogenic influence	Sparse woodland, other woodlands, medium to high cover grassland, lakes, bare soils, saline soils
4	Moderately strong anthropogenic influence	Low-coverage grassland, reservoir ponds, paddy fields, rivers, and canals
5	Stronger anthropogenic influence	Dry field
6	Very strong anthropogenic influence	Rural residential land
7	Excessively strong anthropogenic influence	Urban land, other building lands

**Table 5 ijerph-20-02844-t005:** Calculation formulas and description of each indicator of RSEI.

Index	Calculation Formula and Parameter Description	
WET	$\begin{array}{c} Wet=a_{1}\rho_{blue}+a_{2}\rho_{green}+a_{3}\rho_{red}+a_{4}\rho_{NIR}+a_{5}\rho_{SWIR1}+a_{7}\rho_{SWIR2} \end{array}$	(1)
	where *ρ_i_* (*i* = 1…5, 7) is the reflectance of each TM/ETM+/OLI band, and *ρ_blue_*, *ρ_green_*, *ρ_red_*, *ρ_NIR_*, *ρ_SWIR*1*_*, *ρ_SWIR*2*_* represent the blue, green, red, near-red, mid-infrared bands 1 and 2, respectively. $a_{1}$ (*i* = 1…5, 7) are the sensor parameters.	
NDVI	$\begin{array}{c} NDVI=\frac{\rho_{NIR}-\rho_{red}}{\rho_{NIR}+\rho_{red}} \end{array}$	(2)
	where *ρ_red_*, *ρ_NIR_* have the same meaning as above.	
LST	$\begin{array}{c} L=gain\times DN+bias \end{array}$	(3)
	$\begin{array}{c} T=K_{2}/\ln\left(K_{1}/L+1\right) \end{array}$	(4)
	$\begin{array}{c} LST=\frac{T}{\left[1+\left(\frac{\lambda T}{\rho }\right)\ln\varepsilon \right]}-273.15 \end{array}$	(5)
	where *L* is the radiation value at the sensor in the thermal infrared band; *T* is the temperature value at the sensor; *DN* is a grayscale value; gain and bias are the gain and bias values for the thermal infrared band; *K*_1_ and *K*_2_ are the calibration parameters. λ is the central wavelength in the thermal infrared band, ρ is a constant, ε is the surface emissivity.	
NDBSI	$\begin{array}{c} SI=\left[\left(\rho_{\mathrm{swir}1}+\rho_{\mathrm{red}}\right)-\left(\rho_{\mathrm{nir}}+\rho_{\mathrm{blue}}\right)/\left(\rho_{\mathrm{swir}1}+\rho_{\mathrm{red}}\right)+\left(\rho_{\mathrm{nir}}+\rho_{\mathrm{blue}}\right)\right] \end{array}$	(6)
	$\begin{array}{c} IBI=\frac{2\rho_{\mathrm{swir}1}/\left(\rho_{\mathrm{swir}1}+\rho_{\mathrm{nir}}\right)-\left[\rho_{\mathrm{nir}}/\left(\rho_{\mathrm{nir}}+\rho_{\mathrm{red}}\right)+\rho_{\mathrm{green}}/\left(\rho_{\mathrm{green}}+\rho_{\mathrm{swir}1}\right)\right]}{2\rho_{\mathrm{swir}1}/\left(\rho_{\mathrm{swir}1}+\rho_{\mathrm{nir}}\right)+\left[\rho_{\mathrm{nir}}/\left(\rho_{nir}+\rho_{\mathrm{red}}\right)+\rho_{\mathrm{green}}/\left(\rho_{\mathrm{green}}+\rho_{\mathrm{swir}1}\right)\right]} \end{array}$	(7)
	$\begin{array}{c} NDBSI=SI+IBI/2 \end{array}$	(8)
	where SI is the bare land index, IBI is the building land index, $\rho_{\mathrm{blue}}$, $\rho_{\mathrm{green}}$, $\rho_{\mathrm{red}}$, $\rho_{\mathrm{nir}}$, $\rho_{\mathrm{swir}1}$ have the same meaning as above.	

**Table 6 ijerph-20-02844-t006:** Geodetector principle and description.

Geodetector	Detection Principle		Parameter Description
Factor detector	$q=1-\frac{\sum_{j=1}^{L}N_{h}\sigma_{h}^{2}}{N\sigma^{2}}=1-\frac{\mathrm{SSW}}{\mathrm{SST}}$	(10)	Where q value indicates the explanatory power of the independent variable on the dependent variable, the larger the q value, the greater the effect of the independent variable *X* on the dependent variable *Y*. *h* = 1, …, *L* denotes the classification of the independent variable *X* and $N_{h}$ is the number of stratifications. SSW is within the sum of squares, N is the number of patches in the whole area, $\sigma_{h}^{2}$ indicates the variance of the number of stratifications *h*. SST is the total sum of squares, $\sigma^{2}$ is the variance of the whole area $\sigma_{h}^{2}$ is the variance of the dependent variable *Y* values (RSEI).
	$\begin{array}{c} \mathrm{SSW}=\overset{L}{\underset{j=1}{\sum }}N_{h}\sigma_{h}^{2},\mathrm{SST}=N\sigma^{2} \end{array}$	(11)	
Interaction detector	$q\left(X_{1},X_{2}\right)<Min\left(q_{1},q_{2}\right)$	Nonlinear attenuation	Where $X_{1}$, $X_{2}$ denote two different impact factors, $q\left(X_{1},X_{2}\right)$ denotes the influence of $X_{1}$ when interacting with $X_{2}$, $q_{1},{\;q}_{2}$ denote the influence of $X_{1}$, $X_{2}$ when acting as a single factor, $Min\left(q_{1},q_{2}\right)$, $Max\left(q_{1},q_{2}\right)$ denotes the one with the smallest or largest of $q_{1},q_{2}$.
	$Min\left(q_{1},q_{2}\right)<q\left(X_{1},X_{2}\right)<Max\left(q_{1},q_{2}\right)$	Single factor nonlinear attenuation	
	$q\left(X_{1},X_{2}\right)>Max\left(q_{1},q_{2}\right)$	Double factor enhancement	
	$q\left(X_{1},X_{2}\right)=q_{1}+q_{2}$	Mutually independent	
	$q\left(X_{1},X_{2}\right)>q_{1}+q_{2}$	Non-linear enhancement	

## Data Availability

Not applicable.

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
