# Peer review of "Monitoring Multi-Scale Ecological Change and Its Potential Drivers in the Economic Zone of the Tianshan Mountains’ Northern Slopes, Xinjiang, China"

_ijerph, 2023, doi:10.3390/ijerph20042844_

Round 1

Reviewer 1 Report

1) the abstract should include the methods used, the results part in the abstract should be summarized.

2)in the introduction line 44, as result of what?

3) In figure 1c it is better to change the boundaries color of the study area to be more visible.

4) Line 117 what are the sources of climatic data used, meteorological stations or remote sensing. It should be mentioned.

5) figure 2 in the pre-processing section reflectance not reflectance.

6) in line 139 the year of Ailian, C. el al. should be added.

7) you should mention why you used the PCA method and what are the benefits form it?

8) why in the Reference citation number 5 is in capital letters.

Reviewer 2 Report

1: Before providing details of the ecological impact connecting it to the historical situation in the study area would provide significance of the results.
2: The paper looks very focused on the study area (which is good), but it also puts the generalization of the proposed method in question. It would help if the authors can add a discussion on the generalization of the proposed methodology.
3: In the introduction section, there are a lot of space issues. The papers were cited without giving space between the last letter and the reference number.
4: The paper focuses on RSEI for the results. If the authors can add a few more metrics it would further improve the work.
5: The data cleaning process especially cloud removal processes can be further improved by following the new signal processing techniques such as deep unfolding prior-based learning methods.
6: Captions are very important and please use them as an opportunity to convey key insights about the figure. Figure 2 provides the proposed framework but unfortunately, the caption is not conveying any insight about the proposed framework. Please update.
7: The discussion on the choice of methods for preparing and processing the data is not appropriately covered. Adding a couple of paragraphs describing the rationale behind selecting particular techniques will improve this work a lot.
8: Section 2.3.1 Degree of human interference requires a complete overhaul. Please rewrite and provide details.
9: A brief definition of WET, NDVI, LST, and NDBSI before explaining the formulas can go a long way in improving the readability of the paper. Please add them.
10: The normalization of the four indicators requires further discussion. Please add information about the different scales. Then highlight how the normalization is performed and its benefits.
11: For tables 3 and 4, please cite the sources of the equations and also describe the Factor detector and interaction detector. The current discussion on these detectors in the table is not enough.
12: Section 3, subsection 3.1, and subsubsection 3.1.1 have no line between them. I would suggest adding a couple of lines providing a brief description of the section for the reader.
13: The results are a natural outcome of the framework addressing the questions targeted in the abstract. It is a good practice to connect the results by referring to the research questions.
14: Please add an appropriate caption for figure 4.
15: If the thresholds used in 3.2.1 are standard, please cite the source otherwise please explain the rationale behind the selection of the threshold values.
16: Figure 6 caption: Please use the capital letter to begin the caption.
17: Please provide the limitations and shortcomings of the proposed method as well. The shortcoming discussed in the paper is not enough. Please add more details about the limitations of the proposed framework, measurement challenges, etc.
18: Section 4 has a lot of editorial issues (spacing, reference citing issues, etc.). Please fix them.
19: A literature review of the same sort of studies undertaken will be very useful for positioning this work. A couple of paragraphs would be enough.
20: Caption of the figure 12 needs editorial fixes. Please also improve the resolution of the figure.

Reviewer 3 Report

The text is written concisely and clearly. The number of variables under analysis is large, which makes reading the text somewhat time-consuming.

The methodology created is complex, but well supported. The description of the mathematical tools used is sometimes complicated and requires careful reading, but it is always coherent and correct.

The authors present a methodology that allows the analysis of different human realities, integrating natural and human factors. The literature review is extensive, adequate and current, supporting the research needs.

The created tool can be great for urban and regional planning that is based on an application that is available and ubiquitous today. The article emphasizes the usefulness of an instrument for determining an Ecological Index by Remote Sensing based on Google Earth.

The authors point out the limitations of the study and point out new clues for future investigations.

The conclusions respond to the working hypotheses defined.

Reviewer 4 Report

Please find the comments.

Round 2

Reviewer 4 Report

Everything seems to be well addressed except the citations which are not kept in journal format, I advise authors to look into it.

For example I have pointed some of those:

it should not be capitalise

JUSTICE, C.O.; TOWNSHEND, J.R.G.; HOLBEN, B.N.; TUCKER, C.J. Analysis of the Phenology of Global Vege- 572 tation Using Meteorological Satellite Data. Int J Remote Sens 1985, 6, 1271–1318, doi:10.1080/01431168508948281

Then please check citation number 72 where no authors name is mentioned add them

72. The Spatio-Temporal NDVI Analysis for Two Different Australian Catchments. In Proceedings of the El Sawah, S. (ed.) MODSIM2019, 23rd International Congress on Modelling and Simulation.; Modelling and Simulation Society of Australia and New Zealand, December 1 2019. 

Check the author pattern:

73. Zhengxing, W.; Chuang, L.; Alfredo, H. From AVHRR-NDVI to MODIS-EVI: Advances in Vegetation Index Re-673 search. Acta ecologica sinica 2003, 23, 979–987.

Author Response

Response: The citation format has been checked and corrected. One citation was removed during the change, so that the original 72 and 73 have been advanced by one position.

Thank you very much for your attention and the reviewers' comments and opinions on our paper. Your comments have been very helpful in improving our manuscript and we thank you again for your comments and suggestions. We have revised the manuscript based on your suggestions. We sincerely hope that this manuscript will eventually be accepted and published in International Journal of Environmental Research and Public Health.

If you have any queries, please don't hesitate to contact me at the address below.

Thank you and best regards.

Yours sincerely,

Lina Tang, E-mail: tanglina@stu.xjnu.edu.cn

Corresponding author: Alimujiang·Kasimu

E-mail: alimkasim@xjnu.edu.cn